# Deep CNN-based detection of cardiac rhythm disorders using PPG signals from wearable devices

**Miray Gunay Bulut**[1], **Sencer Unal**[2], **Mohamed Hammad**[3,4], **Paweł Pławiak**[5,6,7]*

**1** Department of Electricity, Malatya Turgut Ozal University, Turkey, **2** Department of Electrical and Electronics Engineering, Firat University, Turkey, **3** EIAS Data Science Lab, College of Computer and Information Sciences, Prince Sultan University, Riyadh, Saudi Arabia, **4** Department of Information Technology, Faculty of Computers and Information, Menoufia University, Shibin El Kom, Egypt, **5** Department of Computer Science, Faculty of Computer Science and Telecommunications, Cracow University of Technology, Krakow, Poland, **6** Institute of Theoretical and Applied Informatics, Polish Academy of Sciences, Gliwice, Poland, **7** Hemolens Diagnostics, Wroclaw, Poland

\* plawiak.pawel@gmail.com

**Data Availability Statement:** This study utilized third party data from the UMMC Simband Dataset for Pulsewatch (https://www.synapse.org/pulsewatch). The authors do not have permission to publicly share minimal data for this study from

## Abstract

Cardiac rhythm disorders can manifest in various ways, such as the heart rate being too fast (tachycardia) or too slow (bradycardia), irregular heartbeats (like atrial fibrillation-AF, ventricular fibrillation-VF), or the initiation of heartbeats in different areas from the norm (extrasystole). Arrhythmias can disrupt the balanced circulation, leading to serious complications like heart attacks, strokes, and sudden death. Medical devices like electrocardiography (ECG) and Holter monitors are commonly used for diagnosing and monitoring cardiac rhythm disorders. However, in recent years, the development of wearable devices has played a significant role in the detection and diagnosis of rhythm disorders through the use of photoplethysmography (PPG) signals. Wearable devices enable patients to continuously monitor their health status and allow doctors to provide earlier diagnoses and interventions. In this study, a 1D-CNN model is proposed to detect arrhythmias using PPG signals. A dataset prepared by the University of Massachusetts Medical Center (UMMC) containing both ECG and PPG signal data was utilized. In this dataset, ECG signals are filtered with a bandpass filter and raw PPG signals are divided into 30-second segments. Accuracy values were obtained by classifying ECG and PPG signals using a 1D CNN model. ECG signals were used as a reference. The proposed model achieved a 95.17% accuracy rate in detecting normal sinus rhythm (NSR), atrial fibrillation (AF), and premature atrial contractions (PAC) from PPG signals. Datasets are available for download on https://www.synapse.org/pulsewatch. The codes used in this study are available on the https://github.com/miraygunay/PPG-Code.git website.

## Introduction

Cardiac arrhythmias are abnormal heart rhythms that can lead to serious health problems such as strokes, heart failure, and even death. Therefore, early detection of arrhythmias is important

UMMC Simband Dataset. Data are available upon request from UMMC representative, Dong Han (Department of Biomedical Engineering, University of Connecticut, Storrs, CT 06269, USA), via email (dong.han@uconn.edu) for researchers. Researchers are required to register a Synapse account (https://accounts.synapse.org/register1), send a request email to Dong Han with your Synapse ID, real name, and organizational affiliation to access these data. The authors confirm that others would be able to access or request these data in the same manner as themselves previously described. The authors also confirm that they did not have any special access or request privileges that others would not have.

**Funding:** The author(s) received no specific funding for this work.

**Competing interests:** The authors have declared that no competing interests exist.

for effective treatment and management [1]. There are many common types of arrhythmias, including Atrial Fibrillation (AF), Premature Atrial Contractions (PACs), and Premature Ventricular Contractions (PVCs). Among these, AF is the most prevalent type of arrhythmia. AF affects millions of people worldwide and is characterized by irregular and rapid heartbeats that can increase the risk of strokes, heart failure, and other cardiovascular complications [2]. Traditionally, electrocardiogram (ECG) signals are used to detect arrhythmias [3]. The use of ECG signals for arrhythmia detection has a long history and well-established methods. ECG signals are obtained by placing electrodes on the skin and measuring the electrical activity of the heart [4]. ECG signals provide high-resolution information about the electrical activity of the heart and are widely used in clinical settings for the diagnosis of arrhythmias. However, ECG signals require the placement of electrodes on the skin, which can be uncomfortable for patients. These signals are often noisy and can be challenging to interpret [5].

An increasingly popular option to overcome the difficulties of ECG in arrhythmia detection is Photoplethysmography (PPG) signals. PPG signals are obtained by measuring changes in light absorption in tissues and allow non-invasive monitoring of the cardiovascular system [6, 7]. The changes in light absorption are due to variations in blood volume in the microvascular bed of tissue, which occur with each heartbeat. These signals are typically collected by shining a light source (usually a Light Emitting Diode (LED) with a wavelength range of 600–1000 nm) onto a part of the body (usually a fingertip or earlobe) and measuring the amount of light absorbed by the underlying tissue. PPG signals consist of a pulsatile waveform that represents changes in blood volume over time. They are often used to monitor heart rate and detect various cardiovascular parameters, including pulse rate, oxygen saturation (SpO2), and pulse waveforms. These signals can be easily obtained through wearable devices such as pulse oximeters, smartwatches, and fitness trackers and are used in various medical and health monitoring applications. Smartwatches, as wearable devices, offer remote health monitoring capabilities using PPG technology. Smartwatches enable the recording and continuous monitoring of PPG signals over extended periods. In this way, the signals received from the heart can be accessed at any time. Various software have been developed to interpret irregular rhythms in these signals, identify characteristics of a normal heartbeat, and analyze the captured data [8–10]. Thanks to this software, heart rhythm irregularities can be easily detected.

Deep learning has emerged as a powerful tool in the field of medical signal processing, particularly suitable for arrhythmia detection due to its ability to learn complex patterns. While deep learning based on Convolutional Neural Networks (CNNs) does not require manual feature engineering, other deep learning approaches can still depend on manual features. Arrhythmias are characterized by complex and often subtle changes in ECG signals. Deep learning models can autonomously identify and analyze complex temporal and spatial patterns present in ECG signals, making them capable of detecting subtle irregularities that might be challenging to manually describe. This approach eliminates the labor-intensive feature extraction process, allowing for more automated and accurate arrhythmia detection while enhancing adaptability to various arrhythmia types. Deep learning has enabled the development of models for arrhythmia detection in both ECG and PPG signals, reducing the need for manual interpretation and improving the speed and accuracy of diagnosis. There are many studies in the literature that focus on using deep learning for AF detection from ECG signals [1, 2, 5, 11–18]. AF detection and classification can be automated using deep learning techniques and algorithms applied to ECG signals. Similarly, deep learning methods are used for AF detection from PPG signals. For instance, CNNs have been applied to the task of AF detection from PPG signals in studies [19, 20]. One advantage of using CNNs for PPG signals is their ability to effectively learn underlying patterns and features in time series data [21–23]. This can help improve the accuracy and robustness of the analysis compared to traditional methods. CNNs can process high-

dimensional data inputs, such as multi-channel PPG signals, and learn complex relationships between inputs and outputs. Additionally, the use of end-to-end deep learning approaches with CNNs can reduce the need for time-consuming and error-prone manual feature engineering. Transfer learning, where a pre-trained CNN is fine-tuned for a specific task using a limited amount of labeled data, has also been found to be effective for processing PPG signals. Studies [9, 19, 20, 24–34], have shown that deep learning algorithms applied to PPG signals can accurately detect these arrhythmias with high sensitivity and specificity. For example, Kwon et al. [32], achieved a 95.85% accuracy in AF and normal sinus rhythm (NSR) detection from PPG signals using a 1D CNN model with a dataset containing 14,298 samples. In another study, Aliamiri et al. [24], developed a convolutional-recurrent hybrid model (CNN-RNN) that could easily classify AF and NSR and achieved a 98.19% accuracy rate by utilizing waveform features in the PPG dataset. As evident from these studies, the high accuracy of deep learning algorithms combined with the non-invasive nature of PPG signals make this approach an appropriate and cost-effective solution for the early detection and management of cardiac arrhythmias.

In this study, a 17-layer 1D-CNN-based model is proposed for the automatic recognition of AF, NSR, and PAC from PPG signals. AF, NSR, and PAC are automatically detected using 30-second segments of PPG signals. The same 17-layer 1D-CNN architecture applied to PPG signals is also applied to the ECG channels within the dataset without any modifications, and the results are compared. The objective of the study is to achieve high-accuracy arrhythmia detection using deep learning methods through wearable technologies with PPG signals. The motivation for this approach is to leverage wearable technology to obtain PPG signals, which simplifies AF detection and addresses the challenges associated with recording ECG signals, while also providing a generalizable 1D CNN model for AF detection from PPG signals.

The contributions of this study are as follows:

- A 1D CNN model has been developed for end-to-end automatic feature extraction and classification from both ECG and PPG signals,

- Introduction of a novel approach using PPG signals and deep learning methods, providing a non-invasive and easily accessible method for continuous monitoring and detection of cardiac arrhythmias,

- The integration of PPG signals, which involve sensors that do not penetrate the skin, contributes to improved patient comfort by addressing concerns related to electrode placement on the skin and promoting a more patient-friendly monitoring experience,

- The performance of the developed 1D CNN model has been evaluated on a dataset comprising both ECG data and signals obtained from a smartwatch, and it has achieved higher results compared to the current literature.

The remaining sections of this study are organized as follows: Related Works section presents recent studies in the literature similar to this research, Materials and Methods section introduces the ECG and PPG datasets and explains in detail the architecture of CNNs used for arrhythmia detection, Experimental Results section presents the results obtained from the model, Discussion section discusses the obtained results, Conclusions section concludes the paper.

## Related works

In the literature, machine learning and deep learning approaches have been employed for disease detection using PPG signals in various studies. For example, Chiang et al. [10], used a support vector machine (SVM) classifier to evaluate the quality of arteriovenous fistulas in hemodialysis patients using a PPG sensor device. In another study, Uçar et al. [35], proposed a

new sleep staging method based on the k-nearest neighbor (KNN) classification algorithm using PPG signals. Kwon et al. [32], investigated 75 patients with atrial fibrillation (AF) who underwent elective direct current cardioversion (DCC). PPG data was collected using a pulse oximeter for 15 minutes before and after DCC. They used deep learning classifiers, specifically 1D-CNN and recurrent neural network (RNN), to detect AF from PPG data by identifying early atrial contractions (PAC) in the presence of sinus rhythm (SR) after successful cardioversion. They achieved AF detection accuracy of 95.85% with 1D-CNN and 96.04% with RNN methods. Ramesh et al. [9], proposed a 1D-CNN neural network using features derived from Heart Rate Variability (HRV) to classify 30-second heart rhythms from PPG data obtained from a smartwatch into NSR and AF categories. They achieved an accuracy of 95.10%, sensitivity of 94.60%, and specificity of 95.20% in the PPG dataset. When applying their model to an ECG dataset, they reached accuracy, sensitivity, and specificity values of 95.50%, 94.50%, and 96.00%, respectively. Shashikumar et al. [20], applied continuous wavelet transform to PPG data obtained from a smartwatch and trained a 1D CNN on the resulting spectrograms to detect AF. The study achieved an AF detection accuracy of 91.8%.

Bashar et al. [36] used PPG signals obtained from smartphones along with an SVM classifier to achieve higher accuracy in AF detection. The proposed method was shown to detect noise artifacts in PPG signals with an accuracy of 91.16% and successfully detect AF. In another study by the same authors [37], they developed a new method to determine whether the PPG signal from a smart wristwatch was corrupted by motion artifacts in addition to AF detection. They detected noise in the PPG signal using a noise detection method based on Variable Frequency Complex Demodulation (VFCDM) and achieved sensitivity, specificity, and accuracy values of 98.18%, 97.43%, and 97.54%, respectively, using a PAC detection algorithm for AF detection. Voisin et al. [19], focused on AF monitoring using PPG signals obtained from a wrist-worn device. They utilized a 50-layer CNN (ResNeXt) and directly took the time-series PPG signal as input to assess the presence or absence of atrial fibrillation. This study highlights the potential of CNN models in detecting atrial fibrillation using PPG signals.

Additionally, Han et al. [38], developed an algorithm for detecting premature atrial contractions (PAC) and premature ventricular contractions (PVC) using PPG data obtained from a smartwatch. Their algorithm combined various techniques, including a novel vector similarity method, to distinguish between different arrhythmias. The results showed promising sensitivity and specificity for PAC/PVC detection. These studies underscore the potential of using PPG signals and CNN models together to detect atrial fibrillation and early contractions. The deep learning approaches used in these studies demonstrate the effectiveness of CNN models in analyzing PPG signals for cardiac arrhythmia detection. These findings contribute to the growing research field of PPG-based detection of cardiovascular diseases and provide insights into the development of accurate and non-invasive diagnostic tools.

## Materials and methods

In this article, arrhythmia detection has been achieved using a deep learning model on PPG signals obtained from wearable devices. For this purpose, a dataset consisting of signals from 37 subjects, including both ECG and PPG signals, was utilized. Preprocessing steps were applied to the PPG and ECG signals in the dataset, including min-max normalization and label encoding. All signals were divided into 30-second lengths without overlap. The ECG signals, filtered with a bandpass filter, and the PPG signals, which were in raw form, were used, as obtained from the data in [38]. A 1D-CNN model capable of effectively operating on the signals within the dataset was designed. The 1D-CNN model was trained and tested on both ECG and PPG signals.

## PPG dataset

The datasets in this study are the original data from the article by Han et al. [38]. Datasets are available for download on https://www.synapse.org/pulsewatch. The researchers conducting the study collected data from 37 patients (28 males and 9 females) with cardiac arrhythmias, aged between 50 and 91, who participated in a smartwatch study at the outpatient cardiovascular clinic of the University of Massachusetts Medical Center (UMMC). ECG and smartwatch data were simultaneously collected using a two-lead rhythm device (Cardea SOLO, Cardiac Insight Inc., Bellevue, WA, USA) and a Samsung Gear S3 smartwatch (Samsung, San Jose, CA, USA) on the chest and wrist, respectively. The ECG probe data consists of single-channel signals sampled at 250 Hz. The Gear S3 data consists of single-channel PPG signals. PPG data from 37 patients were obtained using the Gear S3. Patients with arrhythmia registered at the University of Massachusetts Medical Center wore a Gear S3 smartwatch and an ECG probe while being allowed to perform their normal activities for 14 consecutive days/24 hours. The data were obtained by charging the smartwatches daily and replacing the ECG device's battery on the 7th day. In total, there are 2614 PPG segments, each containing 30 seconds of data sampled at a rate of 50 Hz, with 1500 samples per segment. Among all ECG and PPG samples, 2222 are NSR, 337 are AF, and 55 are PAC. Some example signals of NSR, AF, and PAC from the dataset used in this study are provided in Fig 1.

## Proposed 1D-CNN model

In the field of biomedical signal processing, CNNs are used for various tasks such as ECG analysis and medical image analysis [39]. PPG signals are widely employed in medical applications for monitoring cardiovascular health and, more recently, in computer vision for human activity recognition and biometrics. Unlike traditional machine learning techniques, 1D-CNNs can automatically extract relevant features directly from raw ECG and PPG signals, eliminating the need for manual feature engineering. This enables the model to learn hierarchical representations at different abstraction levels, potentially improving classification accuracy. In this study, a 17-layer 1D-CNN model was designed for the classification of AF, NSR, and PAC arrhythmia types from ECG and PPG data. Fig 2 provides a block representation of the 1D-CNN model used in the study. The model consists of multiple convolutional layers with filters designed to capture specific temporal patterns in the input signals. Each layer in the CNN extracts increasingly complex features from the input signal, allowing the model to learn hierarchical representations that are essential for effective signal classification. Furthermore, the 1D-CNN employs elements such as filters, pooling layers, and activation functions to enhance its ability to recognize distinctive patterns in the PPG data. Through the training process, the model optimizes the weights and biases to minimize classification errors and increase overall performance. The filter numbers are set as 128, 64, 64, 128, 128, 256, 256, and 512. These filters slide over the input data, detecting local features through convolutions. Subsequent pooling layers reduce the dimensionality of the features while preserving relevant information. Table 1 provides a detailed presentation of the parameters of the proposed 17-layer 1D-CNN model.

## Experimental results

### Experimental setup

The methods tested in this study were developed using the Python language in the Google Colab environment. The ECG and PPG datasets, along with the classes they contain, have an equal number of data points and were split into training (70%), validation (15%), and testing (15%) sets, as shown in Table 2. The model was trained over 30 epochs. Some

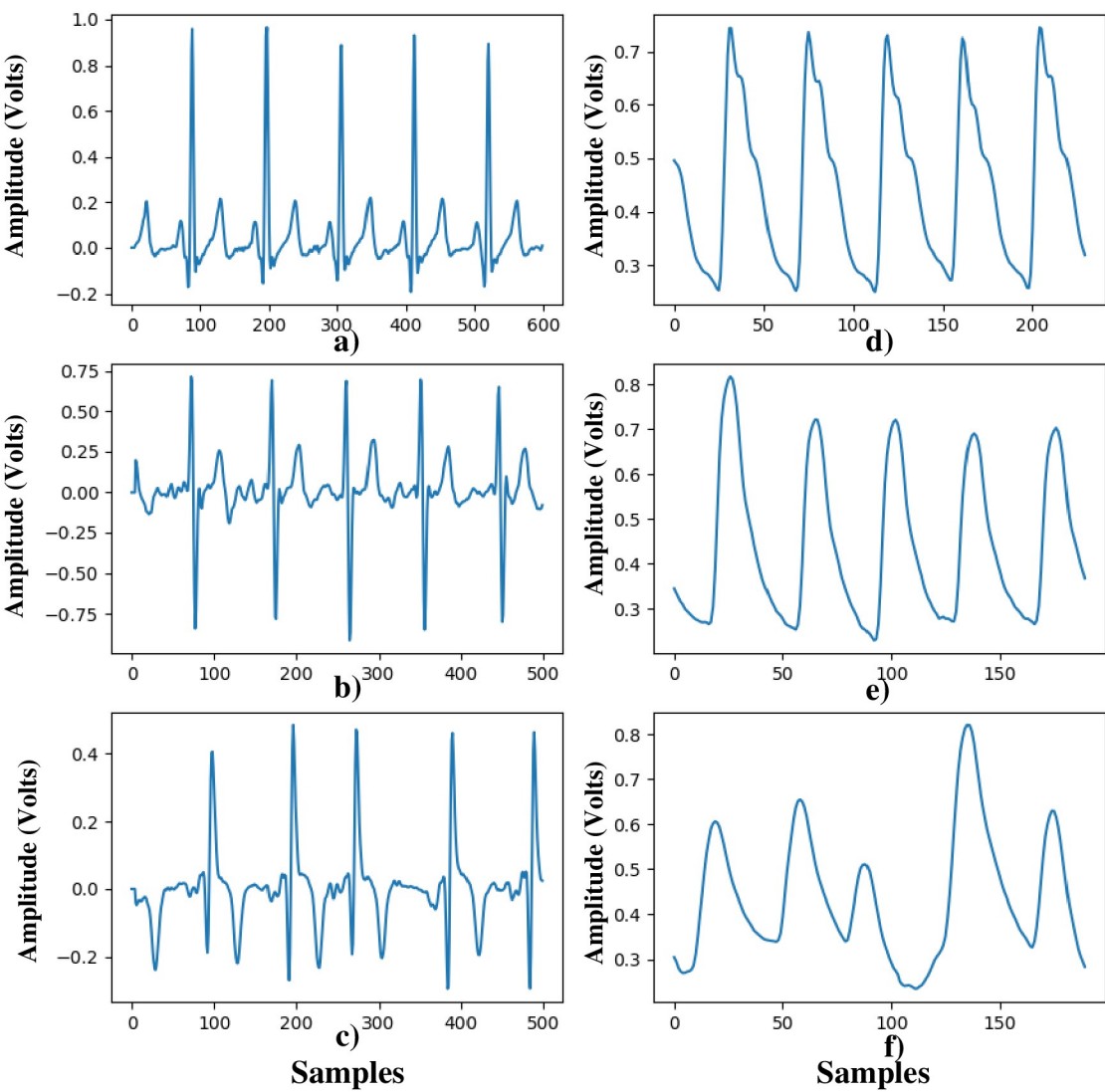

**Fig 1. Representations of a) NSR, b) AF, and c) PAC signals in the ECG dataset, along with d) NSR, e) AF, and f) PAC signals in the PPG dataset.**

hyperparameters of the model include a batch size of 128, the "Adam" optimizer, and the loss function "categorical cross-entropy". The 1D CNN model was trained on labeled datasets consisting of annotated ECG and PPG records with corresponding arrhythmia types. To evaluate the classification performance, common metrics such as accuracy (AC), sensitivity (SEN), specificity (SPE), precision, recall and F1 score were used. Precision represents the model's ability to correctly identify positive samples (true positives) among all predicted positive samples. Sensitivity refers to the proportion of correctly predicted observations in a class among all observations belonging to that class. In other words, it measures the classifier's ability to correctly identify data that is truly positive [40]. The F1 score is the harmonic mean of precision and recall, balancing both measurements and considering both false positives and false negatives.

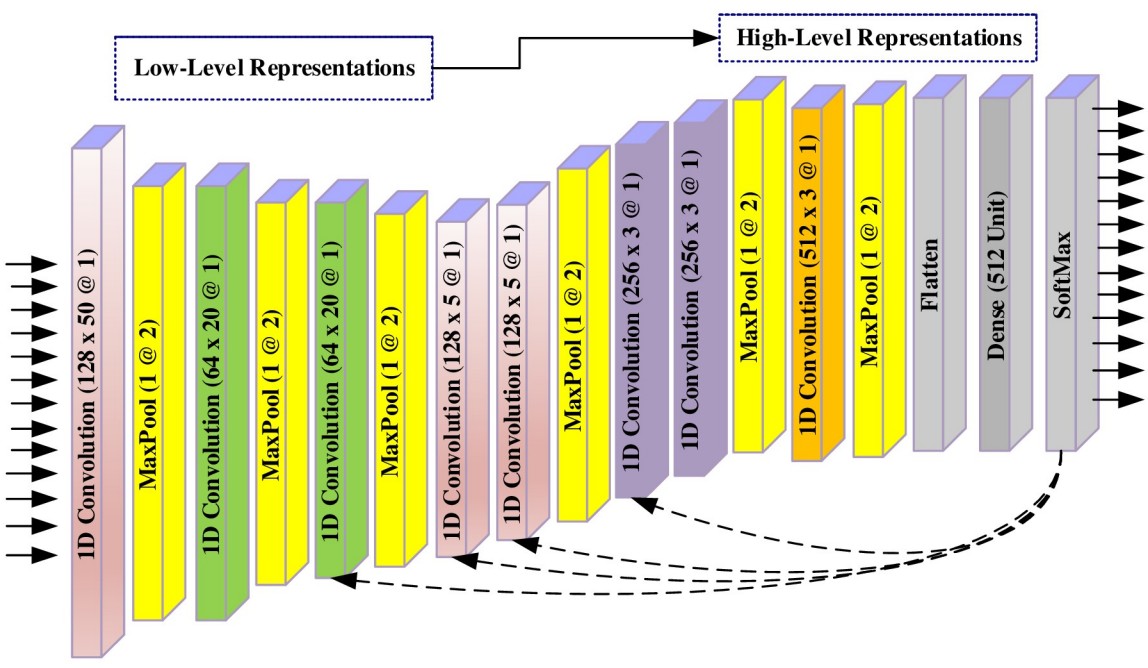

**Fig 2. The architecture of the 17-layer 1D CNN model.**

## Classification results

In this study, the training of the 1D-CNN model used for the classification of PPG and ECG data utilized 1500-dimensional PPG signals with 2614 samples and 3750-dimensional 2-channel ECG data. The model was separately run for PPG and ECG, and the results were obtained. Fig 3 illustrates the accuracy and loss plots of the 1D-CNN model during the training process on PPG signals.

**Table 1. Parameters of the proposed 17-layer 1D CNN model.**

| Layer | Layer Name | Kernel Size | Number of Filters | Other Layer Parameters |
|---|---|---|---|---|
| 1 | Conv1D | 50 | 128 | Activation = ReLU, Strides = 1 |
| 2 | MaxPooling1D | - | - | Pooling Size = 1, Strides = 2 |
| 3 | Conv1D | 20 | 64 | Activation = ReLU, Strides = 1 |
| 4 | MaxPooling1D | - | - | Pooling Size = 1, Strides = 2 |
| 5 | Conv1D | 20 | 64 | Activation = ReLU, Strides = 1 |
| 6 | MaxPooling1D | - | - | Pooling Size = 1, Strides = 2 |
| 7 | Conv1D | 5 | 128 | Activation = ReLU, Strides = 1 |
| 8 | Conv1D | 5 | 128 | Activation = ReLU, Strides = 1 |
| 9 | MaxPooling1D | - | - | Pooling Size = 1, Strides = 2 |
| 10 | Conv1D | 3 | 256 | Activation = ReLU, Strides = 1 |
| 11 | Conv1D | 3 | 256 | Activation = ReLU, Strides = 1 |
| 12 | MaxPooling1D | - | - | Pooling Size = 1, Strides = 2 |
| 13 | Conv1D | 3 | 512 | Activation = ReLU, Strides = 1 |
| 14 | MaxPooling1D | - | - | Pooling Size = 1, Strides = 2 |
| 15 | Flatten | - | - | - |
| 16 | Dense | - | 512 | ReLU |
| 17 | SoftMax | - | - | Softmax |

**Table 2. Number of signal fragments used for ECG and PPG classes.**

| No | Class | Fragment Numbers | Number of Used Fragments | | |
|---|---|---|---|---|---|
| | | | Training | Testing | Validation |
| 1 | Normal sinus rhythm | 2222 | 1555 | 334 | 333 |
| 2 | Atrial fibrillation | 337 | 235 | 51 | 51 |
| 3 | PrematureAtrial contraction | 55 | 39 | 8 | 8 |
| | Total | 2614 | 1829 | 393 | 392 |

The accuracy rate, which started around 84% during the training process, increased to over 94% over 30 epochs. Additionally, the loss value, which was 0.60, decreased to 0.2 after 30 epochs. As can be seen from the accuracy and loss graphs, there was no overfitting during the training stage. Similarly, the performance graphs of the 1D-CNN model for the 2-channel ECG data during the training stage are shown in Fig 4.

When considering the training graphs of the model on ECG signals, it has been observed that there is a quite good training performance for ECG channel 1 data (Fig 4(a)). On the other hand, during the training phase on ECG channel 2 data, an issue of overfitting was observed. The hyperparameters of the model were not modified for PPG, ECG (Channel 1), and ECG (Channel 2) signals. Comparative graphs of the model's performance during the training phase are provided in Fig 5.

The idea that different classification results from signals coming from different ECG channels may be due to electrode position. Each lead captures the heart's electrical activity from a specific viewpoint, resulting in different ECG morphologies. For example, while one lead captures electrical activity from a particular part of the heart, another lead collects data from a completely different perspective. This diversity leads to each lead having different signal characteristics.

Changes in electrode placement cause these signal characteristics to vary, which can affect the outcomes of automatic classification algorithms. When algorithms evaluate these different signal characteristics, they use the unique perspective provided by each lead for classification. Therefore, the different classification results obtained from two ECG channels stem from the differences in electrode positions and the unique perspectives they provide on the heart's electrical activity.

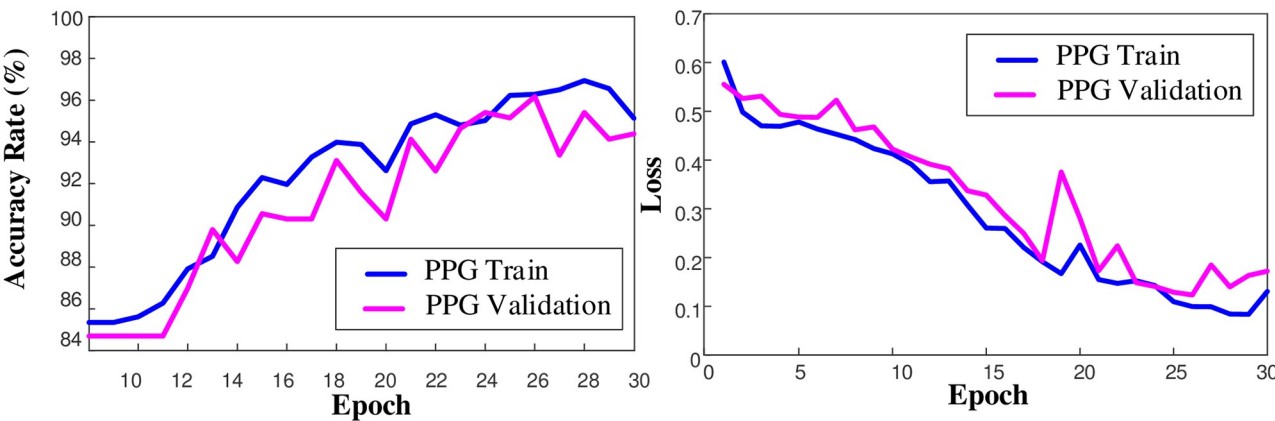

**Fig 3. Accuracy and loss plots for the model trained on the PPG dataset on the training and validation sets.**

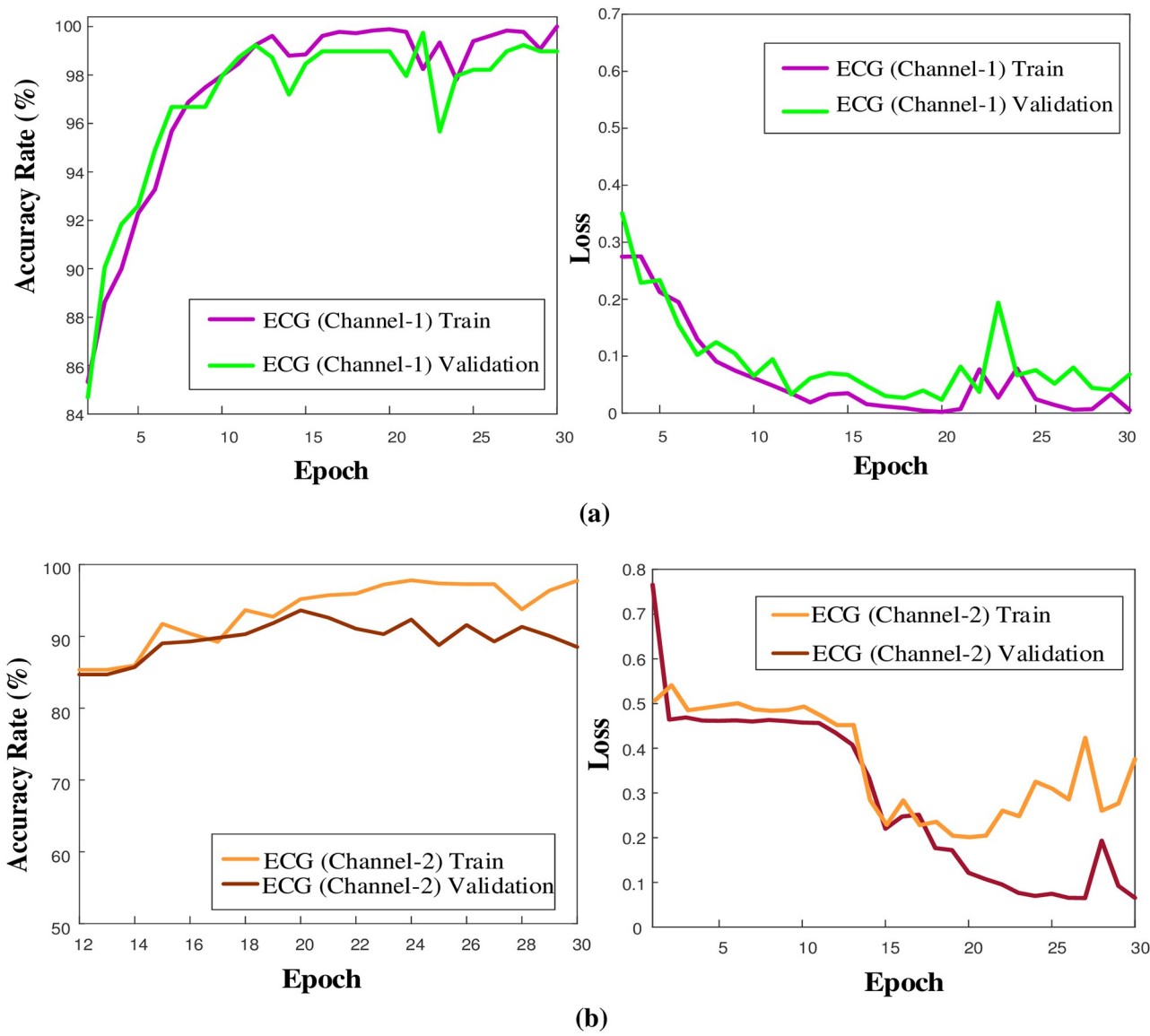

**Fig 4. Training plots of the 1D-CNN model on ECG channel data a) ECG (Channel-1) and b) ECG (Channel-2).**

In summary, the positioning of electrodes can directly influence the morphology of ECG signals and, consequently, the results of classification algorithms. This highlights the critical importance of electrode placement for accurate and consistent ECG analyses.

When evaluating the training performance of the model on PPG and ECG signals during the training phase, the best performance was achieved with ECG channel 1 data. Training using PPG signals yielded relatively lower performance compared to ECG channel 1 input but performed quite well compared to channel 2 input. Following the completion of the training phase, the trained models were assessed on test data. In the test phase, a total of 393 data points were used, with an equal number for both PPG and ECG. The classification of this test data was performed using the trained model, and confusion matrices were obtained, which were then evaluated using various metrics. Fig 6 displays the confusion matrices obtained by the models on both PPG and ECG test data. Table 3 presents the class-based performance values

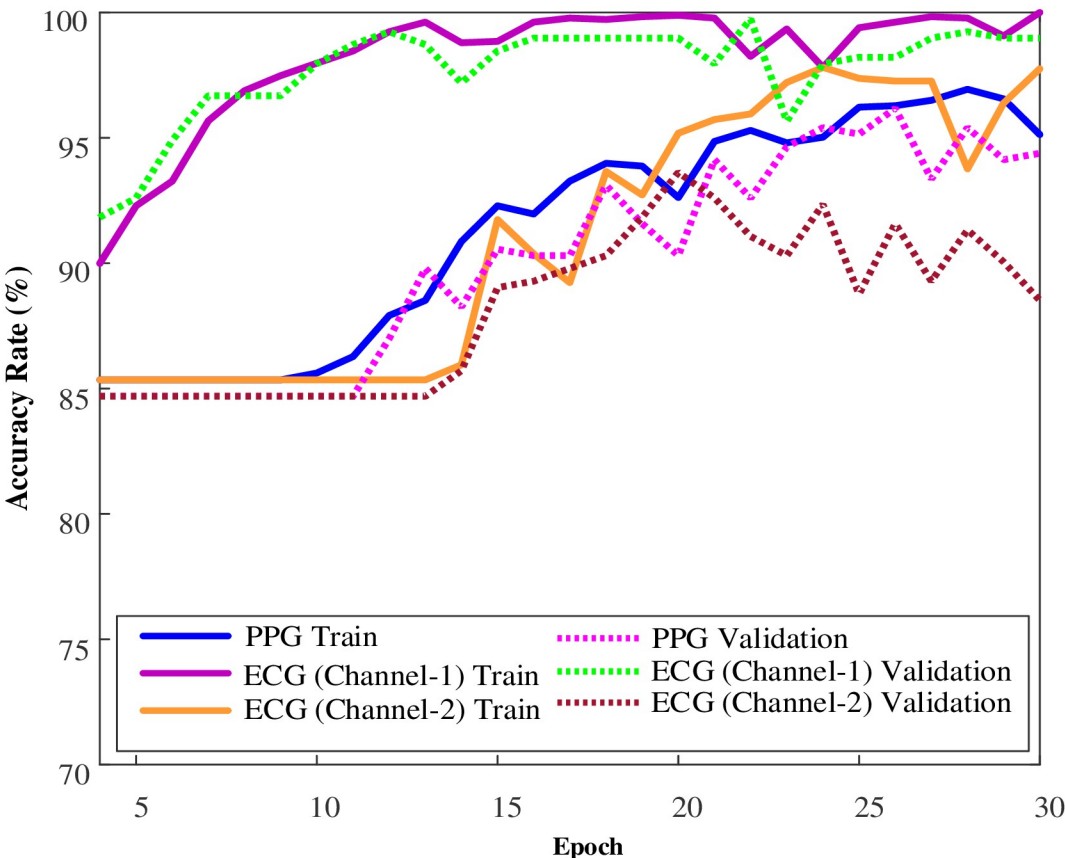

**Fig 5. The comparative graphs of the performance of the model trained using PPG, ECG (Channel-1), and ECG (Channel-2) signals during the training phase.**

of the model for the test data based on common evaluation metrics derived from the confusion matrices.

The PPG dataset achieved an overall accuracy of 95.17%. For the AF (Atrial Fibrillation) class in the PPG dataset, the precision, recall, and F1 score are 0.93, 0.86, and 0.89, respectively. These measurements indicate that the model performs well in detecting AF data with relatively high precision and recall. The NSR class demonstrates accurate classification of NSR samples, achieving sensitivity, specificity, and F1 scores of 0.96, 0.98, and 0.97, respectively. However, the PAC class shows relatively poor performance with low precision, recall, and F1 scores, indicating difficulties in correctly identifying PAC data. The macro-average F1 score for the PPG dataset is 0.74, which is the average of F1 scores across all classes with equal weights. The weighted average F1 score, considering the number of examples in each class during calculation, is 0.95.

From the confusion matrix provided in Fig 6, we can see that there are 50 true positive predictions for AF, 322 true positive predictions for NSR, and 2 true positive predictions for PAC. In the ECG (Channel-1) dataset, the overall accuracy rate is 99.49%, which is higher compared to the PPG input data. The PAC class performs well with precision, recall, and F1 scores of 100%, 83%, and 91%, respectively, indicating accurate identification of PAC samples.

Finally, the overall accuracy of the ECG (Channel-2) dataset is lower at 89.82% compared to the other two datasets. The AF class obtains precision, recall, and F1 scores of 0.84, 0.71,

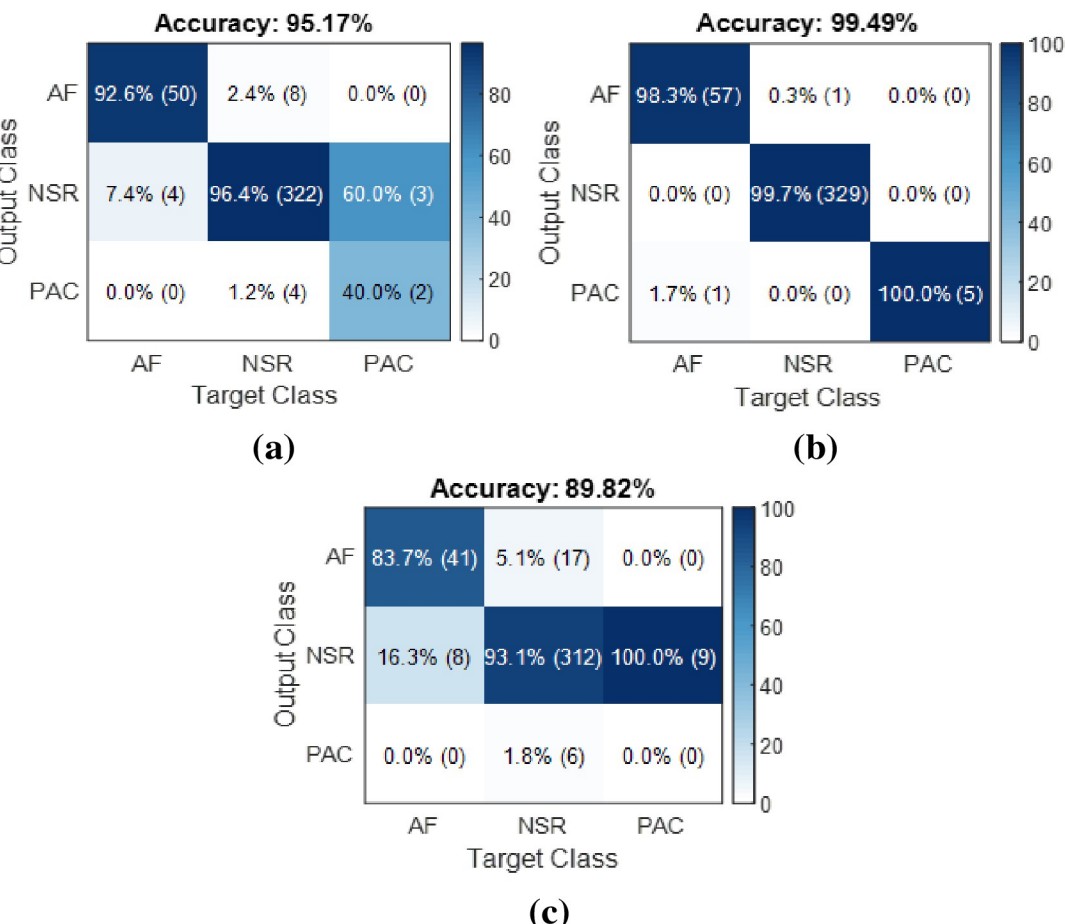

**Fig 6. The confusion matrices obtained by the recommended 1D-CNN model on the test data for a) PPG, b) ECG (Channel-1), and c) ECG (Channel-2).**

and 0.77, respectively, which suggests relatively lower performance in identifying AF samples compared to the other datasets. The model performs poorly on the PAC class for ECG (Channel-2) input. High-performance values are achieved for detecting the AF class compared to other classes within the datasets. Therefore, the most favorable results in terms of classification

**Table 3. The performance metric values of the proposed model for PPG, ECG (Channel-1), and ECG (Channel-2) test data.**

| Input | Class | Accuracy (%) | Sensitivity (%) | Specificity (%) | Precision (%) | F1-Score (%) | Accuracy (%) |
|---|---|---|---|---|---|---|---|
| **PPG** | AF | 86.21 | 86.20 | 98.78 | 93 | 89 | 95.17 |
| | NSR | 98 | 97.87 | 81.25 | 96 | 97 | |
| | PAC | 33.33 | 33 | 99.2 | 40 | 36 | |
| **ECG (Channel-1)** | AF | 98.27 | 98 | 99.70 | 98 | 98 | 99.49 |
| | NSR | 100 | 100 | 98.41 | 100 | 100 | |
| | PAC | 83.33 | 83 | 100 | 100 | 91 | |
| **ECG (Channel-2)** | AF | 71 | 70.69 | 97.5 | 84 | 77 | 89.82 |
| | NSR | 97.50 | 94.83 | 64.06 | 93 | 94 | |
| | PAC | 0 | 0 | 97.51 | 0 | 0 | |

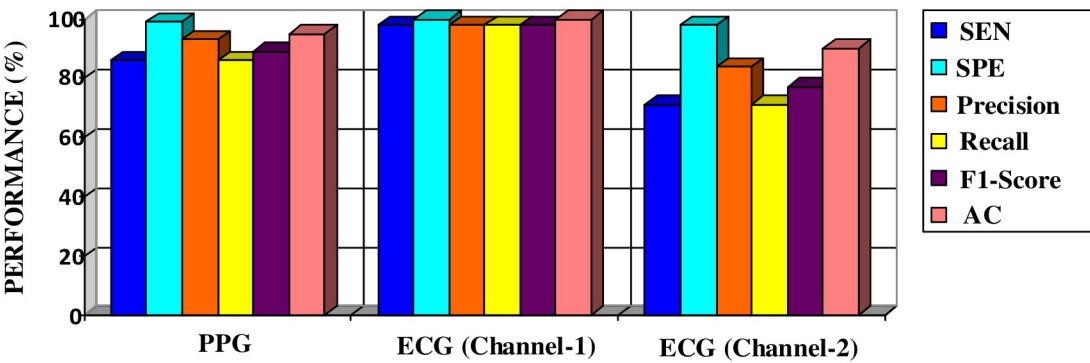

**Fig 7. The performance values of the AF class obtained through the classification of datasets.**

performance are obtained for the AF class. Fig 7 illustrates performance comparisons for the AF class with PPG and ECG input signals.

The results obtained indicate that the model exhibits high performance in classifying with ECG (Channel-1) input. The achievements of the model on PPG signals appear to be relatively lower. However, a high success rate of 95.17% has been achieved for PPG signals.

## Discussion

In this study, a PPG dataset prepared by the University of Massachusetts Medical Center (UMMC) was utilized. Table 4 provides a summary of a comparison of studies developed for the detection of cardiovascular diseases from signals obtained from wearable devices with machine learning methods. Lemay et al. [31], who concentrated on the analysis of PPG signals acquired through a device worn on the wrist, determined 10-second segments as the frame duration and performed feature extraction from these segments.

The extracted features included the average, standard deviation, minimum, and interquartile ranges of cardiac intervals (RR intervals) within a period. Using a support vector machine (SVM) classifier, they identified NSR and AF classes from the extracted features. With their proposed approach, they achieved a classification accuracy of 93.85% for PPG data. Shashikumar et al. [26], obtained PPG signals using the Samsung Simband. They performed image-based feature extraction using deep convolutional neural networks on 10-minute data

**Table 4. A comparison of studies developed for the detection of cardiovascular diseases from signals obtained from wearable devices with machine learning methods.**

| Author | Length of PPG Segments | Measurement Device | Feature Set | Methodology | Performance Results (AC) |
|---|---|---|---|---|---|
| Nemati et al. [30] | 3,5–8,5 Minute | Wrist-worn device Samsung Simband | RR times series features+ sample entropy + standard deviation | Elastic Net logistic model | PPG = %95.00 |
| Bonomi et al. [29] | 120 Second | Wrist-worn device Philips Cardio | RR times series features+ sample entropy+standard deviation+RMSSD | Derived threshold values of features for best ROC | PPG = %98.1 |
| Lemay et al. [31] | 10 Second | Wrist-worn device | RR times series features, mean, minimum, and median | Support Vector Machines (SVM) | PPG = %93.85 |
| Fallet et al. [33] | 10 Second | Wrist-worn device | RR times series features and PPG waveform features | Decision trees | PPG = %95.00 |
| Shashikumar et al. [26] | 30 Second | Wrist-worn device Samsung Simband | R-R Intervals and waveform features | Bidirectional Recurrent Neural Network (BRNN) | ECG = %94.00 PPG = %95.00 |
| **This work** | **30 Second** | **Wrist-worn device Samsung Gear +ECG Device** | **ECG Segment PPG Segment** | **Convolutional Neural Network (CNN)** | **ECG = %99.49 PPG = %95.17** |

segments with 30-second recording windows. They classified the extracted features using a bidirectional recurrent neural network (BRNN) for AF detection. Their designed algorithm achieved a 95% accuracy rate.

In this study, ECG signals obtained from two channels of the electrocardiogram were used for comparison with PPG signals. 30-second segments of the signals considered in the study were used. The developed 17-layer 1D CNN model provided an accuracy rate of 95.17% for PPG signals. This deep learning model effectively captured complex patterns and subtle changes in the input signals, providing improved classification accuracy. Only preprocessing steps such as min-max normalization and label encoding were performed on the PPG and ECG signals. The use of PPG signals in the detection of AF, NSR, and PAC allows for real-time arrhythmia detection, contributing to early diagnosis and management of cardiovascular conditions.

The advantages of this study can be listed as follows:

- The physiological signals in the heart can be easily received with optical sensors using wearable devices, and these devices provide continuous health monitoring.

- Implementation of a 1D CNN deep learning approach in the classification of PPG signals.

- Automatic detection of AF, PAC, and NSR using 30-second PPG segments.

- Early arrhythmia detection and potential for timely medical intervention and improvement in outcomes through PPG signal analysis.

The disadvantages of this study are as follows:

- Limited and imbalanced data distribution (insufficiency of data in the PAC class).

By ensuring that both ECG (Channel-1) and ECG (Channel-2) signals use the same CNN model, the obtained results have been compared with signals obtained from PPG. Accuracy values of 99.49% and 89.82% were obtained from ECG (Channel-1) and ECG (Channel-2), respectively. The accuracy value of 97.17% obtained from PPG is a good value when compared to the two channels of ECG. However, a low accuracy rate has been obtained for the PAC class in the classification of PPG signals, indicating that the model struggles to accurately identify and classify samples belonging to the PAC class. The reason for this is the limited amount of data in the PAC class and the imbalanced data distribution. Imbalanced data has made it difficult for the CNN to effectively learn the specific patterns and features of the PAC class, resulting in low performance in the minority PAC class.

In this study, different classification results were obtained from the signals obtained from both channels of the ECG. In particular, in standard 12-lead ECGs, the electrodes are placed on the limbs (right arm, left arm, right leg, left leg) and chest (V1-V6). Each electrode placement captures electrical signals from various angles, providing a comprehensive view of the heart's activity. The morphology of the ECG waves (P wave, QRS complex, and T wave) changes depending on the electrode position and reflects different aspects of the heart's electrical function. The 2-lead ECG used in this study is a simple type of ECG that usually uses three electrodes to record the electrical activity of the heart. This type of ECG is used for more basic and rapid assessments. In 2-lead ECGs, the electrodes are usually placed on the chest or limbs and provide less information than the 12-lead ECG. Considering these aspects, since the electrode positions of the two channels were different in our study, the signals obtained from the two channels were also different. That is, different electrode positions led to different ECG morphology. In this way, different signals were obtained in our study and different classification results were obtained and the results were compared.

Achieving high accuracy in arrhythmia detection using PPG signals obtained from wearable devices offers numerous advantages; it supports remote monitoring, patient comfort, diagnosis with optical sensors, early detection of disease, and low cost-effectiveness. Moreover, real-time feedback increases the potential for long-term monitoring and wider access to health information. In contrast, the use of ECG signals comes with disadvantages such as complexity and hardware requirements, contact and discomfort, limited long-term monitoring, susceptibility to artifacts, and localization difficulties. ECG provides more accurate cardiological analysis by directly measuring the electrical activity of the heart, but even portable ECG devices have limited ease of use and are less practical for continuous monitoring. In contrast, PPG signals can be easily monitored with wearable devices at home, at work, or during sports, making PPG more accessible for continuous health monitoring and use in daily life. However, the use of wearable devices also presents important limitations and challenges such as signal quality and reliability. In addition to these limitations, factors such as age and sex of the individuals using wearable devices may also affect PPG signals.

In our current study, it was observed that the population was unbalanced in terms of sex (28 males, 9 females). In particular, it has been stated in various studies that thinner skin structure in women may affect PPG signals [41]. Neglecting this biometric factor may limit the performance of our model on women. The age range of the population used in our study (between 51–90) is also an important factor. The incidence of AF increases with age and age-related changes can be observed in PPG signals. Especially in elderly individuals, stiffening of peripheral vessels and changes in blood flow dynamics may affect the quality of PPG signals [41]. In addition, although the incidence of AF is higher in men, this difference decreases with increasing age and causes convergence between the sexes [42].

Despite these, advanced filtering techniques, calibration processes, adaptive algorithms and sensor improvements reduce these problems and increase PPG signal quality and reliability. Advanced deep learning algorithms have the ability to recognize and adapt to differences in sex and age-related data. This means that if the model is trained on a larger and more balanced population in the future, it can provide effective results for users of all ages and sexes. Indeed, the 17-layer CNN model used in our study can be optimized to better analyze sex and age-related differences. Validation of the model to be developed on more balanced populations will ensure that this application can be applied to a wider audience from a clinical perspective. In addition, the ability to monitor remotely and continuously with portable devices increases the potential of PPG signals in arrhythmia detection. Smartwatches can collect uninterrupted data during users' daily activities, and this data offers the advantage of providing continuous health monitoring without affecting the users' quality of life. In conclusion, the proposed 1D CNN model has significant potential to be integrated into the clinical workflow, but considering the population imbalance, sex and age-based biometric differences should be more represented in the model training. Future studies will include validation of the model on more balanced and larger populations, thus increasing its clinical applicability.

In this study, we performed arrhythmia detection using a CNN model applied to PPG signals obtained from the chest and wrist regions. Our results indicate that our designed model can successfully detect arrhythmia using data from these regions. However, it is important to evaluate the adaptability of PPG signals from different body regions to our model for the future expansion and application of our study. Studies focusing on the impact of PPG signals from different areas such as the finger and ear on model performance could enhance its generalization ability and expand its potential for clinical application. This is because PPG signals from different body regions can vary due to physiological and anatomical differences such as tissue density, blood flow, and environmental factors. Regions like the finger and ear may present significant differences in signal quality and patterns.

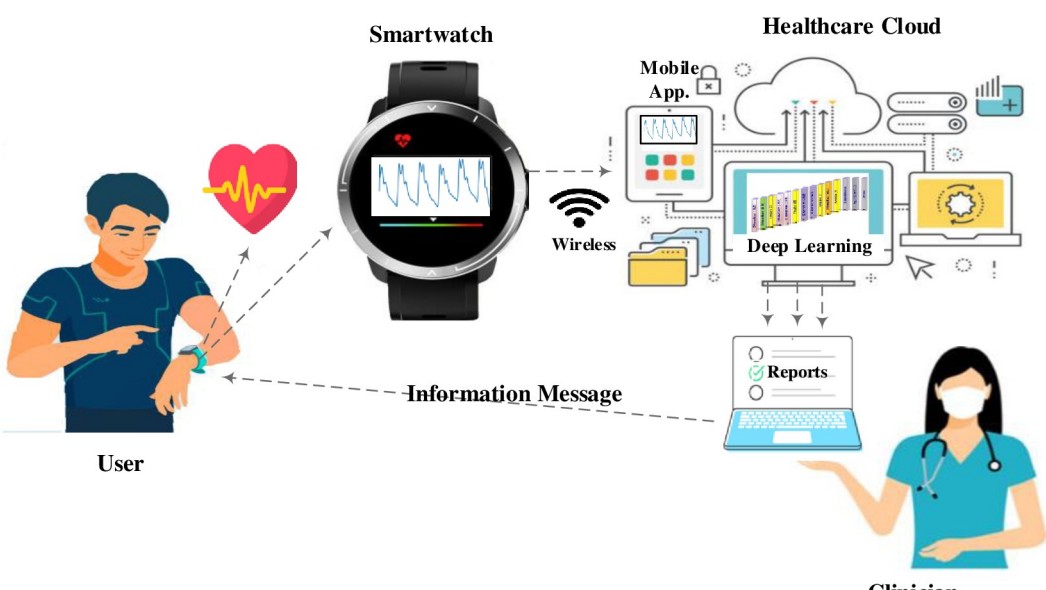

**Fig 8. An example of AF detection scenario created through integration of the smartwatch application into the clinical workflow.**

The findings from our study suggest avenues for further research into how obtaining PPG signals from different body regions may influence model performance. Such advanced studies could improve the reliability of PPG-based arrhythmia detection methods and broaden their utility in clinical applications. In our future work, assessing the adaptability of PPG signals from different body regions to our CNN model holds potential for enhancing arrhythmia detection performance. This will help us better understand the effectiveness and generalizability of these methods in clinical practice.

As a result, our study highlights the potential of our accurate and automated physiological signal analysis approach, not only in terms of the accuracy rates achieved for PPG signals but also for the successful detection of specific cardiac conditions. The advantages of using multimodal data and wearable devices in real-time monitoring further enhance the practicality and relevance of our work. Through this study, future research can build upon these findings by expanding the dataset, considering different populations, and exploring the integration of our approach into clinical workflows, ultimately benefiting patient care and cardiovascular health management. As seen in Fig 8, the proposed solution reduces computational complexity, making it more suitable for real-time signal processing on wearable devices and cloud computing. Additionally, efficient execution, low-latency processing, scalability, and cost-effectiveness enable the application of the solution in these contexts.

The detection of AF using a deep learning model from PPG signals obtained from wearable devices has significant clinical implications and real-world applications. The proposed model in our study is not only suitable for integration into the healthcare system for remote patient monitoring but also highly practical when integrated into wearable devices. The suggested model has the potential to revolutionize healthcare by providing early and continuous AF detection in a non-invasive and convenient manner, reducing the risk of more severe cardiac complications through timely interventions. Additionally, facilitating remote patient monitoring allows individuals with a history of AF or those at risk to be continuously monitored in their daily lives, leading to better condition management and potentially reducing healthcare

costs associated with AF-related hospitalizations. Overall, the integration of this deep learning-based AF detection model into wearable devices represents a promising advancement in healthcare, improving patient outcomes and quality of life.

## Conclusions

Taking heart signals from patients using an electrocardiogram requires patients to visit a healthcare facility. Wearable devices can overcome this challenge by continuously recording and monitoring signals from the heart using Photoplethysmography technology, allowing us to gain insights into heart health wherever we are. Furthermore, PPG signals can be transferred to a healthcare cloud system. These transferred signals are processed using an advanced deep learning model to detect heart disorders, which are then communicated to a specialist. The specialist can provide necessary assessments and transmit information to the patient's phone or smartwatch. Furthermore, the integration of the model into wearable devices is also possible, and this application is highly practical and beneficial. This integration can improve patient outcomes and reduce healthcare costs by enabling the management of AF and other cardiac conditions. In this study, we developed a 1D CNN model focused on detecting AF, NSR, and PAC from PPG signals collected from a smartwatch, achieving an impressive 95% accuracy rate, rivaling other studies in the literature. Additionally, to compare results obtained from PPG signals, we utilized ECG (Channel-1) and ECG (Channel-2) signals. By training the data from these channels in the same CNN model, we obtained accuracy rates of 99% and 90%, respectively.

## Author Contributions

**Conceptualization:** Miray Gunay Bulut, Sencer Unal.

**Data curation:** Miray Gunay Bulut.

**Formal analysis:** Miray Gunay Bulut.

**Investigation:** Miray Gunay Bulut, Mohamed Hammad.

**Methodology:** Miray Gunay Bulut.

**Resources:** Miray Gunay Bulut, Sencer Unal.

**Software:** Miray Gunay Bulut.

**Supervision:** Sencer Unal, Paweł Pławiak.

**Validation:** Miray Gunay Bulut, Sencer Unal, Mohamed Hammad, Paweł Pławiak.

**Visualization:** Miray Gunay Bulut.

**Writing – original draft:** Miray Gunay Bulut, Paweł Pławiak.

**Writing – review & editing:** Miray Gunay Bulut, Sencer Unal, Mohamed Hammad.

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
