## [Decision Letter · Decision Letter 0]

14 Jun 2024

PONE-D-24-02099Deep CNN-Based Detection of Cardiac Rhythm Disorders Using PPG Signals from Wearable DevicesPLOS ONE

Dear Dr. Pławiak,

Thank you for submitting your manuscript to PLOS ONE. After careful consideration, we feel that it has merit but does not fully meet PLOS ONE’s publication criteria as it currently stands. Therefore, we invite you to submit a revised version of the manuscript that addresses the points raised during the review process.

We look forward to receiving your revised manuscript.

Kind regards,

Agnese Sbrollini

Academic Editor

PLOS ONE

Journal Requirements:

4. In the online submission form, you indicated that [The datasets in this study were used with permission from the authors of the article [38].]. 

Reviewers' comments:

Reviewer's Responses to Questions

**Comments to the Author**

1. Is the manuscript technically sound, and do the data support the conclusions?

Reviewer #1: Yes

Reviewer #2: Partly

2. Has the statistical analysis been performed appropriately and rigorously? 

Reviewer #1: No

Reviewer #2: Yes

3. Have the authors made all data underlying the findings in their manuscript fully available?

Reviewer #1: Yes

Reviewer #2: No

4. Is the manuscript presented in an intelligible fashion and written in standard English?

Reviewer #1: No

Reviewer #2: No

5. Review Comments to the Author

Reviewer #1: The paper proposes a 17-layer deep learning model to classify photoplethysmogram (PPG) signals into three classes: Normal sinus rhythm, Atrial fibrillation, and premature atrial contraction. The model was trained and validated using data from wearable devices. For comparison, classification was also conducted using 1-channel and 2-channel ECG data. The 1-channel ECG yielded the best classification results, evaluated based on accuracy, sensitivity, specificity, precision, and F1 score. The performance of PPG was inferior, with the 2-channel ECG producing the least favourable results. The paper presents promising findings; however, it requires major revision in the following areas:

• The structure of the paper is confusing. In the introduction, sections are used to discuss literature, which should instead be part of the related work. A similar issue occurs in the Discussion, where the authors review literature instead of solely comparing results.

• The language used in the paper, particularly in the introduction, is unclear and needs rephrasing and better use of conjunctions.

• It is not clear how the 2-channel ECG were used as an input.

• Additional points of ambiguity will be listed in the comments below.

Comments:

1. Line 51: Traditionally, electrocardiogram (ECG) [3]

Comment: The placement of the reference should be moved to later in the sentence.

2. Line 64: These signals are typically collected by shining a light source (usually an LED)

Comment: LED is an abbreviation that should be defined before used, and it is better to list the wavelength used.

3. Line 79: Deep learning has emerged as a powerful tool in the field of medical signal processing, particularly suitable for arrhythmia detection due to its ability to learn complex patterns and reduce the need for manual feature engineering.

Comment: Deep learning based on Convolutional neural networks does not require manual features; however deep learning can depend on manual features.

4. Line 118: The same model is also applied to the ECG channels within the dataset,

Comment: when the authors say the same model, do they mean the same architecture or after performing fine tuning?

5. Line 132: The integration of PPG signals and the non-invasive nature of the approach:

Comment: PPG is a non-invasive signal what do the author mean by this sentence?

6. Line 146: In the literature, various studies have employed deep learning approaches for disease detection using PPG signals. For example, Chiang et al. [10], used a support vector machine (SVM) classifier to evaluate the quality of arteriovenous fistulas in haemodialysis patients using a PPG sensor device. In another study, Uçar et al. [35], proposed a new sleep staging method based on the k-nearest neighbour (KNN) classification algorithm using PPG signals. Kwon et al.

Comment: SVM and KNN are not deep learning methods, but machine learning ones.

7. Line 222: Fig 1. Representation of ECG signals in the dataset: a) AF, b) NSR, and c) PAC with representation of PPG signals in the dataset: d) AF e) NSR, and f) PAC.

Comment: needs to be rephrased.

8. Line 240: features from the input signal, al-lowing the model.

Comment: A typo in “al-lowing”.

9. Line 242: CNN employs parameters such as filters, pooling layers, and activation functions to enhance its ability to recognize distinctive patterns in the PPG data. Through the training process, the model optimizes these parameters to minimize classification errors.

Comment: filters, pooling layers, and activation functions are not parameters that optimized through the training process as these parameters do not change its value after the training initialized.

10. In Figure 2 numbers of Kernal size and numbers of filters are not easy to interpret, maybe the authors find another way to express it.

11. In Tabel 1: have two separated columns for kernel size and unit and rename the unit to number of filters.

12. Line 261: including label encoding, one-hot encoding,

Comment: are these two the same thing? Then why it is mentioned twice?

13. Line 261: feature scaling (min-max normalization)

Comment: do the authors mean scaling the signal?

14. Line 269: The 1D CNN model was trained on labelled datasets consisting of annotated ECG and PPG records with corresponding arrhythmia types.

Comment: why mention the data here again where the paragraph was discussing the data earlier?

15. Line 274: Sensitivity, also known as recall, measures the model's ability to correctly identify positive samples among all true positive examples.

Comment: The definition needs to be rephrased as this is not accurate.

16. Line 352: The confusion matrix shows the number of true positives, false positives, true negatives, and false negatives for each class:

Comment: Why defining the confusion matrix here where it was introduced earlier in line 328?

17. Line 376: In this study, a PPG dataset prepared by the University of Massachusetts Medical Center (UMMC) was utilized. Table 4 provides a summary of studies focusing on the analysis of PPG signals. Lemay et al. [27],

Comment: the moving to the table of summary is weird and require rephrasing.

18. Line 395: achieved an impressive accuracy rate of 95.17% for PPG signals.

Comment: It is not appropriate to call your results impressive.

19. Line 398: single-hot encoding was performed.

Comment: Do the authors mean “One-hot” why replacing One by single?

20. Line 402: The advantages of this study can be listed as follows:

Comment: The advantages keep mentioned early, although the dataset does not specify how early in stages were the patients at, also listing “Limited and imbalanced data distribution” as an advantage is not understandable.

Reviewer #2: In this manuscript a 17-layers 1D-CNN was trained and validated for the detection of cardiac rhythm disorders using PPG from wearable devices. The proposed model achieved a 95.17% accuracy rate in detecting normal sinus rhythm, atrial fibrillation, and premature atrial contractions from PPG signals.

1. Introduction:

1) The sentence ‘PPG signals enable recording and continuous monitoring over extended periods, providing access to heart signals at any time’ at line 73-75 is not clear. Can Authors reformulate it?

2) The abbreviation NSR was note introduced before its use. Can Authors introduce it where they write normal sinus rhythm for the first time?

3.1. PPG Data Set:

3) The population is sex-unbalanced (28 male vs 9 female). The incidence of atrial fibrillation is greater in males than in females, but this gap closes with advancing age (population in the present study aged between 51 and 90). Further sex-related differences in photoplethysmography signals exist. I suggest to train and validate the proposed 1D CNN on a sex-balanced population. At least, Authors should deeply discuss this limitation in the discussion.

4) The acquisition protocol is not clear. Are the subjects in a laboratory or in a daily life situation? How much did the recordings last? Can Authors add the description of setting and timing of acquisition?

5) Are subjects undergoing any cardiac therapy?

6) In figure 1, are the PPG windows corresponding to the ECG windows? In the PPG each pulse corresponds to a cardiac beat, thus if there are 2 cardiac beats in the ECG I expect two pulses in the PPG window. Can Authors check and correct the figure?

3.2. Proposed 1D-CNN Model:

7) In line 240 ‘al-lowing’ should be allowing.

4.1. Experimental result:

8) This section contains methodological information, I suggest to move it in the method chapter.

9) In the list of metrics, line 272, both sensitivity and recall appear. Authors should select only one metric name and use it consistently along the paper.

4.2. Classification results:

10) The two ECG channels contains information of the same cardiac electrical activity in corresponding windows but they provide different results. May it depend on electrode position? Can Authors discuss this result in detail and elucidate on the different electrode position and ECG morphology in the two channels?

5. Discussion:

11) In the discussion section Authors should discuss more in deep their results and limitations. Further Authors should provide a more comprehensive comparison with literature.

12) Can Authors discuss the adaptability of the presented CNN to PPG signals acquired in different body locations (ex. Finger, wrist, ear..)?

13) Authors continuously compare the ECG and the PPG. Especially in lines 429 – 432 there is a list of ECG disadvantages “the use of ECG signals comes with disadvantages such as complexity and hardware requirements, contact and discomfort, limited long-term monitoring, invasive electrode placement, sensitivity to artifacts, and localization difficulties.” Although, the superficial ECG is non-invasive and the discomfort of wet electrodes has been overcome by dry electrodes and capacitive sensors for ECG, which are incorporated in wearable devices and everyday usable garments and objects. The PPG recording is also sensitive to artifacts. Can Authors provide a more realistic and balanced comparison of wearable devices for ECG and PPG acquisition systems?

6. PLOS authors have the option to publish the peer review history of their article (what does this mean?). If published, this will include your full peer review and any attached files.

Reviewer #1: No

Reviewer #2: No

---

## [Author Response · Author response to Decision Letter 0]

28 Aug 2024

Manuscript Ref. No: PONE-D-24-02099

Title of Paper : Deep CNN model for AF and PAC detection with PPG signals from Wearable Devices

Authors: Miray Gunay Bulut, Sencer Unal, Mohamed Hammad, Paweł Pławiak

First of all, we would like to thank the reviewers and the editorial board for their valuable remarks and constructive comments on the manuscript. Remarks and comments suggested by reviewers have been applied in the manuscript and responses to reviewers are also given in detail. The revised form of the manuscript is uploaded through the journal web page.

We have made a substantial revision of our paper (the revised parts are marked in red), and provided detailed and itemized responses to the comments of reviewers.

AUTHOR’S RESPONSE TO COMMENTS OF THE REVIEWER#1

The paper presents promising findings; however, it requires major revision in the following areas:

• The structure of the paper is confusing. In the introduction, sections are used to discuss literature, which should instead be part of the related work. A similar issue occurs in the Discussion, where the authors review literature instead of solely comparing results.

We would like to thank to the reviewer for this valuable contribution. Based on the comments made, we added more detailed information about the results to the “Discussion” section.

• The language used in the paper, particularly in the introduction, is unclear and needs rephrasing and better use of conjunctions.

Based on the comments, the article was reviewed and revised.

• It is not clear how the 2-channel ECG were used as an input.

Based on the comments, we added additional information about ECG to the “PPG data set section”.

• Additional points of ambiguity will be listed in the comments below.

1-Comment of Reviewer1: 1. Line 51: Traditionally, electrocardiogram (ECG) [3]

Comment: The placement of the reference should be moved to later in the sentence.

Author’s Response

We would like to thank to the reviewer for this valuable contribution. According to this comment, we corrected the sentence in the “Introduction” section as follows.

 “…Traditionally, electrocardiogram (ECG) signals are used to detect arrhythmias [3].”

2-Comment of Reviewer1: 2. Line 64: These signals are typically collected by shining a light source (usually an LED)

Comment: LED is an abbreviation that should be defined before used, and it is better to list the wavelength used.

Author’s Response

We would like to the reviewer for this contribution. According to this comment we added following sentences to “Introduction” section.

“…These signals are typically collected by shining a light source (usually a Light Emitting Diode (LED) with a wavelength range of 600-1000 nm) onto a part of the body (usually a fingertip or earlobe) and measuring the amount of light absorbed by the underlying tissue.”

3-Comment of Reviewer1: 3. Line 79: Deep learning has emerged as a powerful tool in the field of medical signal processing, particularly suitable for arrhythmia detection due to its ability to learn complex patterns and reduce the need for manual feature engineering.

Comment: Deep learning based on Convolutional neural networks does not require manual features; however deep learning can depend on manual features.

Author’s Response

We would like to the reviewer for this contribution. According to this comment, we corrected the sentence in the “Introduction” section as follows.

“...Deep learning has emerged as a powerful tool in the field of medical signal processing, particularly suitable for arrhythmia detection due to its ability to learn complex patterns. While deep learning based on Convolutional Neural Networks (CNNs) does not require manual feature engineering, other deep learning approaches can still depend on manual features.”

4-Comment of Reviewer1: 4. Line 118: The same model is also applied to the ECG channels within the dataset, Comment: when the authors say the same model, do they mean the same architecture or after performing fine tuning?

Author’s Response

We would like to thank to the reviewer for this valuable contribution. According to this comment, we corrected the sentence in the “Introduction” section as follows.

 “…The same 17-layer 1D-CNN architecture applied to PPG signals is also applied to the ECG channels within the dataset without any modifications, and the results are compared.”

5-Comment of Reviewer1: 5. Line 132: The integration of PPG signals and the non-invasive nature of the approach:

Comment: PPG is a non-invasive signal what do the author mean by this sentence?

Author’s Response

We would like to thank to the reviewer for this valuable contribution. According to this comment we added following sentences to “Introduction” section.

 “…The integration of PPG signals, which involve sensors that do not penetrate the skin, contributes to improved patient comfort by addressing concerns related to electrode placement on the skin and promoting a more patient-friendly monitoring experience.”

6-Comment of Reviewer1: 6. Line 146: In the literature, various studies have employed deep learning approaches for disease detection using PPG signals. For example, Chiang et al. [10], used a support vector machine (SVM) classifier to evaluate the quality of arteriovenous fistulas in haemodialysis patients using a PPG sensor device. In another study, Uçar et al. [35], proposed a new sleep staging method based on the k-nearest neighbour (KNN) classification algorithm using PPG signals. Kwon et al.

Comment: SVM and KNN are not deep learning methods, but machine learning ones.

Author’s Response

We would like to thank to the reviewer for this valuable contribution. According to this comment, we corrected the sentence in the “Related Works” section as follows.

 “…In the literature, machine learning and deep learning approaches have been employed for disease detection using PPG signals in various studies.”

7-Comment of Reviewer1: 7. Line 222: Fig 1. Representation of ECG signals in the dataset: a) AF, b) NSR, and c) PAC with representation of PPG signals in the dataset: d) AF e) NSR, and f) PAC.

Comment: needs to be rephrased.

Author’s Response

We would like to thank to the reviewer for this valuable contribution. According to this comment, we corrected the sentence in the “PPG data set” section as follows.

 “…Fig 1. Representations of a) NSR, b) AF, and c) PAC signals in the ECG dataset, along with d) NSR, e) AF, and f) PAC signals in the PPG dataset.”

8-Comment of Reviewer1: 8. Line 240: features from the input signal, al-lowing the model.

Comment: A typo in “al-lowing”.

Author’s Response

We would like to thank to the reviewer for this valuable contribution. According to this comment, we corrected the sentence in the “Proposed 1D-CNN model” section as follows.

 “…Each layer in the CNN extracts increasingly complex features from the input signal, allowing the model to learn hierarchical representations that are essential for effective signal classification.”

9-Comment of Reviewer1: 9. Line 242: CNN employs parameters such as filters, pooling layers, and activation functions to enhance its ability to recognize distinctive patterns in the PPG data. Through the training process, the model optimizes these parameters to minimize classification errors.

Comment: filters, pooling layers, and activation functions are not parameters that optimized through the training process as these parameters do not change its value after the training initialized.

Author’s Response

We would like to thank to the reviewer for this valuable contribution. According to this comment, we corrected the sentence in the “Proposed 1D-CNN model” section as follows.

 “…Furthermore, the 1D CNN employs elements such as filters, pooling layers, and activation functions to enhance its ability to recognize distinctive patterns in the PPG data. Through the training process, the model optimizes the weights and biases to minimize classification errors and increase overall performance.”

10-Comment of Reviewer1: In Figure 2 numbers of Kernal size and numbers of filters are not easy to interpret, maybe the authors find another way to express it.

Author’s Response

We would like to thank to the reviewer for this valuable contribution. According to this comments we edited Figure 2 as follows in the “Proposed 1D-CNN model” section.

 “… 

11-Comment of Reviewer1: In Table 1: have two separated columns for kernel size and unit and rename the unit to number of filters.

Author’s Response

We would like to thank to the reviewer for this valuable contribution. According to this comments we edited Table 1 as follows in the “Proposed 1D-CNN model” section.

 “

Layer Layer Name Kernel Size Number of Filters Other Layer Parameters

1 Conv1D 50 128 Activation = ReLU, Strides = 1

2 MaxPooling1D - - Pooling Size = 1, Strides = 2

3 Conv1D 20 64 Activation = ReLU, Strides = 1

4 MaxPooling1D - - Pooling Size = 1, Strides = 2

5 Conv1D 20 64 Activation = ReLU, Strides = 1

6 MaxPooling1D - - Pooling Size = 1, Strides = 2

7 Conv1D 5 128 Activation = ReLU, Strides = 1

8 Conv1D 5 128 Activation = ReLU, Strides = 1

9 MaxPooling1D - - Pooling Size = 1, Strides = 2

10 Conv1D 3 256 Activation = ReLU, Strides = 1

11 Conv1D 3 256 Activation = ReLU, Strides = 1

12 MaxPooling1D - - Pooling Size = 1, Strides = 2

13 Conv1D 3 512 Activation = ReLU, Strides = 1

14 MaxPooling1D - - Pooling Size = 1, Strides = 2

15 Flatten - - -

16 Dense - 512 ReLU

17 SoftMax - - Softmax

12-Comment of Reviewer1: 12. Line 261: including label encoding, one-hot encoding,

Comment: are these two the same thing? Then why it is mentioned twice?

Author’s Response

We would like to thank to the reviewer for this valuable contribution. 

Label Encoding: Converts categorical data into numerical values. For example, it transforms categories like "red," "green," and "blue" into numerical values such as 1, 2, and 3, respectively.

One-Hot Encoding: Converts each category into binary vectors. Each category is represented by a vector with a 1 in the position corresponding to that category and 0s elsewhere. For example, the categories "red," "green," and "blue" can be encoded as follows:

"red" -> [1, 0, 0]

"green" -> [0, 1, 0]

"blue" -> [0, 0, 1]

These two methods offer different advantages and disadvantages for various situations. Therefore, label encoding and one-hot encoding are not the same and are distinct encoding methods.

According to this comment we added following sentences to “Experimental Setup” section.

 “…Preprocessing steps were applied to the PPG and ECG signals in the dataset, including min-max normalization and label encoding.”

13-Comment of Reviewer1: 13. Line 261: feature scaling (min-max normalization)

Comment: do the authors mean scaling the signal?

Author’s Response

We would like to thank to the reviewer for this valuable contribution. 

 In our code, the line “scaler = MinMaxScaler(feature_range=(0, 1))” scales the signal. This line performs an operation that transforms data features into a specific range (typically [0, 1]). In other words, it rescales the values of data points according to a predefined min-max range.

14-Comment of Reviewer1: 14. Line 269: The 1D CNN model was trained on labelled datasets consisting of annotated ECG and PPG records with corresponding arrhythmia types.

Comment: why mention the data here again where the paragraph was discussing the data earlier?

Author’s Response

We would like to thank to the reviewer for this valuable contribution. We removed this sentence from the “Experimental Setup” section:“The 1D CNN model was trained on labeled datasets consisting of annotated ECG and PPG records with corresponding arrhythmia types”

15-Comment of Reviewer1: 15. Line 274: Sensitivity, also known as recall, measures the model's ability to correctly identify positive samples among all true positive examples.

Comment: The definition needs to be rephrased as this is not accurate.

Author’s Response

We would like to thank to the reviewer for this valuable contribution. According to this comment we added following sentences to “Experimental Setup” section.

 “…Sensitivity refers to the proportion of correctly predicted observations in a class among all observations belonging to that class. In other words, it measures the classifier's ability to correctly identify data that is truly positive.”

16-Comment of Reviewer1: 16. Line 352: The confusion matrix shows the number of true positives, false positives, true negatives, and false negatives for each class:

Comment: Why defining the confusion matrix here where it was introduced earlier in line 328?

Author’s Response

We would like to thank to the reviewer for this valuable contribution. According to this comment we removed this sentence from the “Classification Results” section: “The confusion matrix shows the number of true positives, false positives, true negatives, and false negatives for each class.”

17-Comment of Reviewer1: 17. Line 376: In this study, a PPG dataset prepared by the University of Massachusetts Medical Center (UMMC) was utilized. Table 4 provides a summary of studies focusing on the analysis of PPG signals. Lemay et al. [27],

Comment: the moving to the table of summary is weird and require rephrasing.

Author’s Response

We would like to thank to the reviewer for this valuable contribution. According to this comment, we corrected the sentence in the “Discussion” section as follows.

 “…Table 4 provides a summary of a comparison of studies developed for the detection of cardiovascular diseases from signals obtained from wearable devices with machine learning methods.”

18-Comment of Reviewer1: 18. Line 395: achieved an impressive accuracy rate of 95.17% for PPG signals.

Comment: It is not appropriate to call your results impressive.

Author’s Response

We would like to thank to the reviewer for this valuable contribution. According to this comment, we changed and rewrote the following sentence in the “Discussion” section.

 “…The developed 17-layer 1D CNN model provided an accuracy rate of 95.17% for PPG signals.”

19-Comment of Reviewer1: 19. Line 398: single-hot encoding was performed.

Comment: Do the authors mean “One-hot” why replacing One by single?

Author’s Response

According to this comment, we changed and rewrote the following sentence in the “Discussion” section.

 “…Only preprocessing steps such as min-max normalization and label encoding were performed on the PPG and ECG signals.”

20-Comment of Reviewer1: 20. Line 402: The advantages of this study can be listed as follows:

Comment: The advantages keep mentioned early, although the dataset does not specify how early in stages were the patients at, also listing “Limited and imbalanced data distribution” as an advantage is not understandable. 

Author’s Response

We would like to thank to the reviewer for this valuable contribution. "Limited and imbalanced data distribution" is given in the disadvantages section of the “Discussion” section. That's why it wasn't changed.

“The disadvantages of this study are as follows:

• Limited and imbalanced data distribution (insufficiency of data in the PAC class).”

Manuscript Ref. No: PONE-D-24-02099

Title of Paper : Deep CNN model for AF and PAC detection with PPG signals from Wearable Devices

Authors: Miray Gunay Bulut, Sencer Unal, Mohamed Hammad, Paweł Pławiak

First of all, we would like to thank the reviewers and the editorial board for their valuable remarks and constructive comments on the manuscript. Remarks and comments suggested by reviewers have been applied in the manuscript and responses to reviewers are also given in detail. The revised form of the manuscript is uploaded through the journal web page.

We have made a substantial revision of our paper (the revised parts are marked in red), and provided detailed and itemized responses to the comments of reviewers.

AUTHOR’S RESPONSE TO COMMENTS OF THE REVIEWER#2

1-Comment of Reviewer2: The sentence ‘PPG signals enable recording and continuo

---

## [Decision Letter · Decision Letter 1]

16 Sep 2024

PONE-D-24-02099R1Deep CNN-Based Detection of Cardiac Rhythm Disorders Using PPG Signals from Wearable DevicesPLOS ONE

Dear Dr. Pławiak,

Thank you for submitting your manuscript to PLOS ONE. After careful consideration, we feel that it has merit but does not fully meet PLOS ONE’s publication criteria as it currently stands. Therefore, we invite you to submit a revised version of the manuscript that addresses the points raised during the review process.

We look forward to receiving your revised manuscript.

Kind regards,

Agnese Sbrollini

Academic Editor

PLOS ONE

Reviewers' comments:

Reviewer's Responses to Questions

**Comments to the Author**

1. If the authors have adequately addressed your comments raised in a previous round of review and you feel that this manuscript is now acceptable for publication, you may indicate that here to bypass the “Comments to the Author” section, enter your conflict of interest statement in the “Confidential to Editor” section, and submit your "Accept" recommendation.

Reviewer #1: All comments have been addressed

Reviewer #2: (No Response)

2. Is the manuscript technically sound, and do the data support the conclusions?

Reviewer #1: Yes

Reviewer #2: Partly

3. Has the statistical analysis been performed appropriately and rigorously? 

Reviewer #1: Yes

Reviewer #2: Yes

4. Have the authors made all data underlying the findings in their manuscript fully available?

Reviewer #1: Yes

Reviewer #2: Yes

5. Is the manuscript presented in an intelligible fashion and written in standard English?

Reviewer #1: Yes

Reviewer #2: Yes

6. Review Comments to the Author

Reviewer #1: The authors successfully revised the paper, addressing all my questions and providing an improved interpretation of the results. However, a minor revision is still required due to the following issues:

1. Typographical errors:

o Line 31: There is a typo in “mod-el”; it should be “model.”

o Line 221: “data set” should be written as “dataset.”

2. Clarification needed on ECG leads:

In line 69, the authors state:

"The 2-lead ECG used in this study is a simple type of electrocardiogram (ECG) that uses two electrodes to record the heart's electrical activity."

While two electrodes can be used for a single-lead ECG (excluding the body reference electrode, however it was not excluded in the describtion of the 12-lead ecg), a two-lead ECG generally involves at least three electrodes. The dataset used by the authors does contain two channels, which does suggest that it is indeed a 2-lead ECG. This explanation, however, needs further clarification to avoid confusion. A correction is recommended.

Reviewer #2: Abstract

1) In the added lines 41-44, “Datasets are available for download on” is repeated twice. Further, the entire sentence is linked, not only the specific links to the repositories.

Introduction

2) The paragraph (lines 64-73) about the difference between 2-leads ECG and 12-leads ECG was inserted in the introduction that is hard to follow now. This paragraph is useful to understand the results of the study, thus authors should move it to the discussion section.

PPG Data Set

3) Can authors add the measurement units on the y-axis label in figure 2?

Discussion

4) Even if future works aim to train and validate their 1D CNN model on a more balanced population in terms of sex and age, the authors should here discuss this crucial aspect regarding the clinical applicability of their work. As commented int the first revision, the population is sex-unbalanced (28 male vs 9 female). The incidence of atrial fibrillation is greater in males than in females, but this gap closes with advancing age (population in the present study aged between 51 and 90). Further sex-related differences in photoplethysmography signals exist.

At the present form, the proposed method is limited and it is not clinically applicable to a general population.

Authors should deeply discuss these limitations in the discussion section.

5) In lines 552-553, authors stated “Signal quality can be affected by motion artifacts, electrode contact issues, and skin variability, making it difficult to obtain accurate PPG signals”. PPG is not acquired by electrodes, can authors correct their statement?

6) What do authors refer to with “invasive electrode placement (For Invasive EKG Devices)”?

Non-invasive Electrocardiographic monitoring can be performed by dry electrodes and capacitive sensors for ECG, which can be incorporated in wearable and portable devices. The authors stated that ECG acquisition require contact with the subject, but the PPG acquisition on the chest or on the wrist through smartwatches also require contact with the device and possibly discomfort.

Can Authors provide a more realistic and balanced comparison of wearable devices for ECG and PPG acquisition systems?

7) In table 4, authors should add ECG segment to the feature set of their work.

7. PLOS authors have the option to publish the peer review history of their article (what does this mean?). If published, this will include your full peer review and any attached files.

Reviewer #1: No

Reviewer #2: No

---

## [Author Response · Author response to Decision Letter 1]

6 Oct 2024

Manuscript Ref. No: PONE-D-24-02099

Title of Paper : Deep CNN model for AF and PAC detection with PPG signals from Wearable Devices

Authors: Miray Gunay Bulut, Sencer Unal, Mohamed Hammad, Paweł Pławiak

First of all, we would like to thank the reviewers and the editorial board for their valuable remarks and constructive comments on the manuscript. Remarks and comments suggested by reviewers have been applied in the manuscript and responses to reviewers are also given in detail. The revised form of the manuscript is uploaded through the journal web page.

We have made a substantial revision of our paper (the revised parts are marked in red), and provided detailed and itemized responses to the comments of reviewers.

AUTHOR’S RESPONSE TO COMMENTS OF THE REVIEWER#1

1-Comment of Reviewer1: 1. Typographical errors:

• Line 31: There is a typo in “mod-el”; it should be “model.”

• Line 221: “data set” should be written as “dataset.”

Author’s Response

We would like to thank to the reviewer for this valuable contribution. According to this comment, we corrected the sentences as follows.

 “In this study, a 1D-CNN model is proposed to detect arrhythmias using PPG signals.”

 “PPG dataset”

2-Comment of Reviewer1: Clarification needed on ECG leads:

In line 69, the authors state:

"The 2-lead ECG used in this study is a simple type of electrocardiogram (ECG) that uses two electrodes to record the heart's electrical activity."

While two electrodes can be used for a single-lead ECG (excluding the body reference electrode, however it was not excluded in the describtion of the 12-lead ecg), a two-lead ECG generally involves at least three electrodes. The dataset used by the authors does contain two channels, which does suggest that it is indeed a 2-lead ECG. This explanation, however, needs further clarification to avoid confusion. A correction is recommended.

Author’s Response

We would like to the reviewer for this contribution. According to this comment, we corrected the sentence in the “Introduction” section as follows. 

“The 2-lead ECG used in this study is a simple type of ECG that usually uses three electrodes to record the electrical activity of the heart. This type of ECG is used for more basic and quick assessments. In 2-lead ECGs, the electrodes are usually placed on the chest or limbs. It provides less information than a 12-lead ECG.”

According to the suggestion of the Reviewer2, this paragraph has been moved to the discussion section.

Manuscript Ref. No: PONE-D-24-02099

Title of Paper : Deep CNN model for AF and PAC detection with PPG signals from Wearable Devices

Authors: Miray Gunay Bulut, Sencer Unal, Mohamed Hammad, Paweł Pławiak

First of all, we would like to thank the reviewers and the editorial board for their valuable remarks and constructive comments on the manuscript. Remarks and comments suggested by reviewers have been applied in the manuscript and responses to reviewers are also given in detail. The revised form of the manuscript is uploaded through the journal web page.

We have made a substantial revision of our paper (the revised parts are marked in red), and provided detailed and itemized responses to the comments of reviewers.

AUTHOR’S RESPONSE TO COMMENTS OF THE REVIEWER#2

1-Comment of Reviewer2: 1) In the added lines 41-44, “Datasets are available for download on” is repeated twice. Further, the entire sentence is linked, not only the specific links to the repositories

Author’s Response

We would like to thank to the reviewer for this valuable contribution. According to this comment, we corrected the sentence in the “Abstract” section as follows.

 “…Datasets are available for download on https://www.synapse.org/pulsewatch."

2-Comment of Reviewer2: The paragraph (lines 64-73) about the difference between 2-leads ECG and 12-leads ECG was inserted in the introduction that is hard to follow now. This paragraph is useful to understand the results of the study, thus authors should move it to the discussion section.

Author’s Response

We would like to thank to the reviewer for this valuable contribution. According to this comment, we moved this paragraph to the “Discussion” section.

 “In this study, different classification results were obtained from the signals obtained from both channels of the ECG. In particular, in standard 12-lead ECGs, the electrodes are placed on the limbs (right arm, left arm, right leg, left leg) and chest (V1-V6). Each electrode placement captures electrical signals from various angles, providing a comprehensive view of the heart's activity. The morphology of the ECG waves (P wave, QRS complex, and T wave) changes depending on the electrode position and reflects different aspects of the heart's electrical function. The 2-lead ECG used in this study is a simple type of ECG that usually uses three electrodes to record the electrical activity of the heart. This type of ECG is used for more basic and rapid assessments. In 2-lead ECGs, the electrodes are usually placed on the chest or limbs and provide less information than the 12-lead ECG. Considering these aspects, since the electrode positions of the two channels were different in our study, the signals obtained from the two channels were also different. That is, different electrode positions led to different ECG morphology. In this way, different signals were obtained in our study and different classification results were obtained and the results were compared.”

3-Comment of Reviewer2: PPG Data Set

Can authors add the measurement units on the y-axis label in figure 1?

Author’s Response

We would like to thank to the reviewer for this valuable contribution. According to this comments we edited Figure 1 as follows in the “PPG dataset” section.

4-Comment of Reviewer2: Discussion

Even if future works aim to train and validate their 1D CNN model on a more balanced population in terms of sex and age, the authors should here discuss this crucial aspect regarding the clinical applicability of their work. As commented in the first revision, the population is sex-unbalanced (28 male vs 9 female). The incidence of atrial fibrillation is greater in males than in females, but this gap closes with advancing age (population in the present study aged between 51 and 90). Further sex-related differences in Photoplethysmography signals exist. At the present form, the proposed method is limited and it is not clinically applicable to a general population. Authors should deeply discuss these limitations in the discussion section.

Author’s Response

We would like to thank to the reviewer for this valuable contribution. According to this comment, we have revised the discussion section and added the following paragraphs.

 “In our current study, it was observed that the population was unbalanced in terms of sex (28 males, 9 females). In particular, it has been stated in various studies that thinner skin structure in women may affect PPG signals [41]. Neglecting this biometric factor may limit the performance of our model on women. The age range of the population used in our study (between 51-90) is also an important factor. The incidence of AF increases with age and age-related changes can be observed in PPG signals. Especially in elderly individuals, stiffening of peripheral vessels and changes in blood flow dynamics may affect the quality of PPG signals [41]. In addition, although the incidence of AF is higher in men, this difference decreases with increasing age and causes convergence between the sexes [42].

Despite these, advanced filtering techniques, calibration processes, adaptive algorithms and sensor improvements reduce these problems and increase PPG signal quality and reliability. Advanced deep learning algorithms have the ability to recognize and adapt to differences in sex and age-related data. This means that if the model is trained on a larger and more balanced population in the future, it can provide effective results for users of all ages and sexes. Indeed, the 17-layer CNN model used in our study can be optimized to better analyze sex and age-related differences. Validation of the model to be developed on more balanced populations will ensure that this application can be applied to a wider audience from a clinical perspective. In addition, the ability to monitor remotely and continuously with portable devices increases the potential of PPG signals in arrhythmia detection. Smartwatches can collect uninterrupted data during users' daily activities, and this data offers the advantage of providing continuous health monitoring without affecting the users' quality of life. In conclusion, the pro-posed 1D CNN model has significant potential to be integrated into the clinical workflow, but considering the population imbalance, sex and age-based biometric differences should be more represented in the model training. Future studies will include validation of the model on more balanced and larger populations, thus increasing its clinical applicability.”

5-Comment of Reviewer2: In lines 552-553, authors stated “Signal quality can be affected by motion artifacts, electrode contact issues, and skin variability, making it difficult to obtain accurate PPG signals”. PPG is not acquired by electrodes, can authors correct their statement?

Author’s Response

We would like to thank to the reviewer for this valuable contribution. This sentence has been changed and combined with the paragraph mentioning the limitations of PPG, which was added in line with comment 4.

“...However, the use of wearable devices also presents important limitations and challenges, such as signal quality and reliability. In addition to these limitations, factors such as the age and gender of people using wearable devices can also affect PPG signals.”

6-Comment of Reviewer2: What do authors refer to with “invasive electrode placement (For Invasive EKG Devices)”? Non-invasive Electrocardiographic monitoring can be performed by dry electrodes and capacitive sensors for ECG, which can be incorporated in wearable and portable devices. The authors stated that ECG acquisition require contact with the subject, but the PPG

acquisition on the chest or on the wrist through smartwatches also require contact with

the device and possibly discomfort. Can Authors provide a more realistic and balanced comparison of wearable devices for ECG and PPG acquisition systems?

Author’s Response

According to this comments, a more proper comparison of ECG and PPG acquisition is presented in the following paragraph.

“Achieving high accuracy in arrhythmia detection using PPG signals obtained from wearable devices offers numerous advantages; it supports remote monitoring, patient comfort, diagnosis with optical sensors, early detection of disease, and low cost-effectiveness. Moreover, real-time feedback increases the potential for long-term monitoring and wider access to health information. In contrast, the use of ECG signals comes with disadvantages such as complexity and hardware requirements, contact and discomfort, limited long-term monitoring, susceptibility to artifacts, and localization difficulties. ECG provides more accurate cardiological analysis by directly measuring the electrical activity of the heart, but even portable ECG devices have limited ease of use and are less practical for continuous monitoring. In contrast, PPG signals can be easily monitored with wearable devices at home, at work, or during sports, making PPG more accessible for continuous health monitoring and use in daily life. However, the use of wearable devices also presents important limitations and challenges such as signal quality and reliability. In addition to these limitations, factors such as age and sex of the individuals using wearable devices may also affect PPG signals.”

7-Comment of Reviewer2: In table 4, authors should add ECG segment to the feature set of their work.

Author’s Response

We would like to thank to the reviewer for this valuable contribution. According to this comment, we added the following sentence to table 4.

“ECG segment”

---

## [Decision Letter · Decision Letter 2]

6 Nov 2024

Deep CNN-Based Detection of Cardiac Rhythm Disorders Using PPG Signals from Wearable Devices

PONE-D-24-02099R2

Dear Dr. Pławiak,

We’re pleased to inform you that your manuscript has been judged scientifically suitable for publication and will be formally accepted for publication once it meets all outstanding technical requirements.

Kind regards,

Agnese Sbrollini

Academic Editor

PLOS ONE

Additional Editor Comments (optional):

Reviewers' comments:

Reviewer's Responses to Questions

**Comments to the Author**

1. If the authors have adequately addressed your comments raised in a previous round of review and you feel that this manuscript is now acceptable for publication, you may indicate that here to bypass the “Comments to the Author” section, enter your conflict of interest statement in the “Confidential to Editor” section, and submit your "Accept" recommendation.

Reviewer #2: All comments have been addressed

2. Is the manuscript technically sound, and do the data support the conclusions?

Reviewer #2: Partly

3. Has the statistical analysis been performed appropriately and rigorously? 

Reviewer #2: Yes

4. Have the authors made all data underlying the findings in their manuscript fully available?

Reviewer #2: Yes

5. Is the manuscript presented in an intelligible fashion and written in standard English?

Reviewer #2: Yes

6. Review Comments to the Author

Reviewer #2: Authors have applied the correction suggested in the previous review. The discussion section was substantially improved.

7. PLOS authors have the option to publish the peer review history of their article (what does this mean?). If published, this will include your full peer review and any attached files.

Reviewer #2: No

---

## [Editor Report · Acceptance letter]

18 Nov 2024

PONE-D-24-02099R2 

PLOS ONE

Dear Dr. Pławiak, 

I'm pleased to inform you that your manuscript has been deemed suitable for publication in PLOS ONE. Congratulations! Your manuscript is now being handed over to our production team.

Kind regards, 

on behalf of

Dr. Agnese Sbrollini 

Academic Editor

PLOS ONE